# The secreted endoribonuclease ENDU-2 from the soma protects germline immortality in *C. elegans*

Wenjing Qi [1✉], Erika D. V. Gromoff[1], Fan Xu[1,2], Qian Zhao[1], Wei Yang[1], Dietmar Pfeifer[3], Wolfgang Maier[1], Lijiang Long[1] & Ralf Baumeister[1,2,4,5]

Multicellular organisms coordinate tissue specific responses to environmental information via both cell-autonomous and non-autonomous mechanisms. In addition to secreted ligands, recent reports implicated release of small RNAs in regulating gene expression across tissue boundaries. Here, we show that the conserved poly-U specific endoribonuclease ENDU-2 in *C. elegans* is secreted from the soma and taken-up by the germline to ensure germline immortality at elevated temperature. ENDU-2 binds to mature mRNAs and negatively regulates mRNA abundance both in the soma and the germline. While ENDU-2 promotes RNA decay in the soma directly via its endoribonuclease activity, ENDU-2 prevents misexpression of soma-specific genes in the germline and preserves germline immortality independent of its RNA-cleavage activity. In summary, our results suggest that the secreted RNase ENDU-2 regulates gene expression across tissue boundaries in response to temperature alterations and contributes to maintenance of stem cell immortality, probably via retaining a stem cell specific program of gene expression.

[1] Bioinformatics and Molecular Genetics (Faculty of Biology), Albert-Ludwigs-University Freiburg, Freiburg, Germany. [2] Spemann Graduate School of Biology and Medicine (SGBM), Albert-Ludwigs-University Freiburg, Freiburg, Germany. [3] Department of Hematology, Oncology and Stem Cell Transplantation, Medical Center-University of Freiburg, Faculty of Medicine, University of Freiburg, Freiburg, Germany. [4] Center for Biochemistry and Molecular Cell Research (ZBMZ, Faculty of Medicine), Albert-Ludwigs-University Freiburg, Freiburg, Germany. [5] Signalling Research Centers BIOSS and CIBSS, Albert-Ludwigs-University Freiburg, Freiburg, Germany. ✉email: wenjing.qi@biologie.uni-freiburg.de

Germ cells are the only type of cells within an organism that deliver genetic and epigenetic materials to the offspring. Germ cells are therefore distinct from the somatic cells in their ability to maintain totipotency for production of an entire organism upon fertilization, and their immortality to allow reproduction for unlimited future generations. In *C. elegans*, loss of germline immortality typically results in steadily increased incidence of sterility of initially fertile animals over generations. To date, telomerase-mediated telomere maintenance, balanced heterochromatic H3K9 methylation, and euchromatic H3K4/H3K36 methylation are the prevalent known molecular mechanisms involved in maintenance of germline immortality[1–4]. Piwi RNA mutants and mutations in multiple components of a nuclear RNA interference (RNAi) pathway that promotes the inheritance of germline RNAi also result in a temperature-dependent mortal germline (Mrt) phenotype[5,6]. Failure to maintain germline immortality in Piwi RNA mutants has recently been explained by transgenerational silencing of histone genes[7]. These studies, however, focused almost exclusively on autonomous mechanisms within the germline. It is unknown whether somatic signaling may contribute to maintenance of germline immortality.

*C. elegans* ENDU-2 belongs to a conserved but less studied family of proteins containing proposed poly-U specific endoribonuclease (XendoU) domains. While human EndoU (PP11 placental protein 11) is used as a cancer marker gene, Viral EndoU Nsp15 is highly conserved in all known coronaviruses and has been suggested to promote viral RNA replication and limit innate immunity response of the host cells[8,9]. *C. elegans* ENDU-2 has been shown to regulate cold stress response[10]. Very recently, ENDU-2 was reported to regulate nucleotides metabolism and germline proliferation in response to alterations in nucleotide levels and genotoxic stresses[11].

Here we report that secretion of the poly-U specific endoribonuclease ENDU-2 from the soma to the germline preserves germline immortality at elevated temperature. We find that ENDU-2 binds to mature mRNAs and downregulates mRNA levels both in the soma and in the germline. In addition, RNA-binding and -cleavage are two separable activities utilized by ENDU-2 to control gene expression via distinct mechanisms. In the soma, ENDU-2 relies on its endoribonuclease activity to negatively regulate a subset of its mRNA targets. In the germline, ENDU-2 prevents misexpression of soma-specific genes and ensures stem cell immortality primarily via its RNA-binding activity, suggesting an essential role of ENDU-2 in retaining the stem cell-specific program of gene expression. In summary, our data suggest that the soma sends ENDU-2 as a messenger to the germline to control gene expression across tissue boundaries in response to temperature alterations.

## Results

**Loss of *endu-2* causes a temperature-dependent Mrt phenotype.** Freshly outcrossed *endu-2(lf)* mutants were phenotypically similar to wild type animals in the first few generations, except displaying slightly reduced germline proliferation, an egg-laying defect (Egl) due to abnormal development of the vulva, and a moderate reduction in adult lifespan at 20 °C (Supplementary Fig. 1). Long-term strain maintenance at 20 °C was difficult, since the *endu-2(lf)* mutants showed gradually increased sterility that, however, could be reset by additional outcrosses with wild type animals. We examined the fertility of two *endu-2* alleles *tm4977* and *by188* over generations after four additional outcrosses. *tm4977* allele has a 620 bp deletion starting from the 5′UTR to the end of the third intron, while the 20 bp deletion in *by188* causes an early stop codon in the first exon. Therefore, these two alleles are probably

null mutants. Both *tm4977* and *by188* alleles became sterile at 20 °C after about 15–20 generations (Fig. 1a, Supplementary Fig. 2a). Since we did not observe a significant reduction of the number of germ nuclei in the mitotic zone across generations, sterility was not the consequence of declining germline proliferation (Supplementary Fig. 2b). This result also indicates that the Mrt phenotype is probably not caused by altered activity of CTPS-1, an ENDU-2 controlled cytidine triphosphate synthase playing an essential role in germline proliferation[11]. Instead, *endu-2* day 1 adult animals in the generations with highly penetrant sterile phenotype showed pleiotropic defects in the germline. The most prominent defects were abnormal cell death (38%, $n = 105$) and increased number of apoptotic corpses (31% with ≥2 corpses per gonad arm, $n = 105$) (Fig. 1b). We also observed prolonged spermatogenesis 24 h after mid-L4 stage (11%, $n = 105$), whereas wild type animals had completely switched spermatogenesis to oogenesis. Furthermore, endomitosis occurred in the mitotic region (14%, $n = 25$) (Fig. 1b). In the generations displaying strong sterile phenotype, *endu-2(lf)* additionally showed high incidence of male progeny (Him, 14%, $n = 218$) and increased occurrence of other phenotypes (Rol, Dpy, Sma phenotypes, in total 10%, $n = 218$). The latter were probably caused by elevated somatic mutation rate, since these phenotypes were not heritable. The sterile phenotype was 100% penetrant at 25 °C already at generation 6–10, but could not be detected at 15 °C (Supplementary Fig. 2c and d). Notably, the highly penetrate sterility could be fully reversed within several generations by transferring the animals to 15 °C (Fig. 1a). Taken together, these results suggest an essential role of ENDU-2 in preserving germline immortality at elevated temperature and the existence of additional mechanisms to compensate for the loss of *endu-2* at 15 °C.

**ENDU-2 is a secreted protein.** A previous report has suggested a wide-spread expression of *endu-2* in somatic tissues[10], but did not report ENDU-2 localization in the germline. To retest the expression pattern of ENDU-2, we generated several independent transcriptional and translational *endu-2* reporters with EGFP as well as a CRISPR/Cas9 EGFP knock-in strain at the endogenous *endu-2* locus (expression constructs are shown in Supplementary Fig. 3a). A transcriptional fusion reporter gene, harboring 4 kb of *endu-2* upstream sequences, was expressed only in the intestine (Supplementary Fig. 3b). A translational reporter harboring the same upstream sequences and the entire genomic region of *endu-2*, fused to EGFP, rescued Egl, reduced germline proliferation, and short lifespan phenotypes (Supplementary Fig. 1a, c and d). The CRISPR/Cas9 EGFP knock-in strain did not display the Mrt phenotype (Supplementary Fig. 1e). These results suggest that ENDU-2::EGFP fusion proteins in the transgenic and the EGFP knock-in strains are functional. We observed ENDU-2::EGFP protein in the cytoplasm of intestine, somatic gonad, and coelomocytes in both *endu-2::EGFP* transgenic and the CRISPR/Cas9 EGFP knock-in strains (Fig. 2a and Supplementary Fig. 3c). Unlike suggested in the study of Ujisawa et al., we did not detect ENDU-2::EGFP in the musculature or in the neurons. Instead, we noticed extracellularly localized ENDU-2::EGFP, such as in the interspace between uterine wall and embryos (Fig. 2a and Supplementary Fig. 3c), suggesting that ENDU-2 may be a secreted protein. ENDU-2 indeed contains a predicted N-terminal (1–19 amino acid) secretion signal peptide for endoplasmic reticulum (ER) targeting, indicating that the protein is destined toward the secretory pathway. Fusion of this secretion signal peptide to the N-terminus of a EGFP reporter gene expressed exclusively in the neurons led to only weak expression level of EGFP protein in the neurons, but an EGFP signal in the extracellular space and the coelomocytes, cells that are known to endocytose secreted

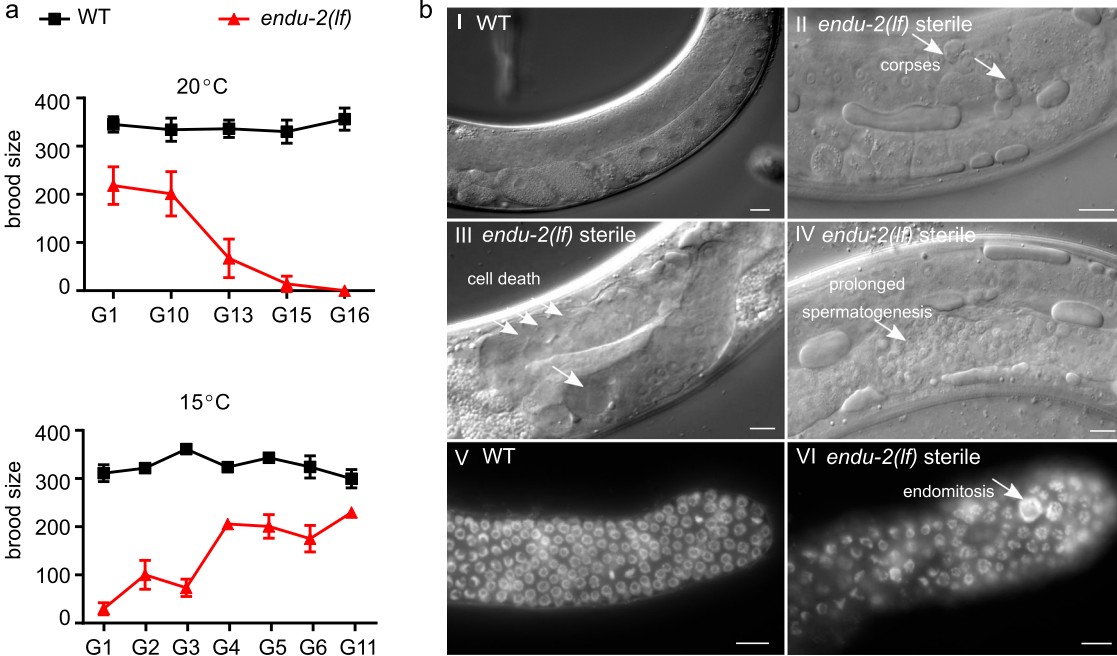

**Fig. 1 endu-2 mutant shows a temperature-dependent Mrt phenotype. a** Upper graph: Brood size of *endu-2(tm4977)*, after 4× initial backcrosses with wild type (G0), decreases over generations at 20 °C. Lower graph: The reproduction defect of *endu-2(tm4977)* animals is reversible by shifting the animals to 15 °C. To obtain this graph, adult animals from G6 that had been maintained at 25 °C were shifted back to 15 °C and counted as G0. As the numbers of generation to reach 100% sterility at 20 °C varied from 10 to 20 among the four biological replicates, only one represented replicate with n = 15 animals for each generation is shown. The data are mean ± SEM. **b** The sterile *endu-2(tm4977)* animals display multiple defects in germline mitosis and meiosis. (I–IV) DIC images of wild type and sterile *endu-2(tm4977)* animals 24 h after mid-L4 stage. The white arrows point to (II) increased number of germline apoptotic corpses; (III) empty gonad due to germ cell death; (IV) prolonged spermatogenesis; (V) DAPI staining of germline proliferating zone of day-one wild type and (VI) sterile *endu-2(tm4977)* adults, respectively. White arrows in VI point to abnormally large chromosomes due to endomitosis. Scale bar 10 μm. N = 3 biological replicates.

proteins from the pseudocoelic fluid (Fig. 2b). In addition, removing the N-terminal secretion signal peptide from ENDU-2 ($\Delta_{ss}$ENDU-2::EGFP) resulted in strong localization of $\Delta_{ss}$ENDU-2::EGFP only in the intestine but not in the coelomocytes (Supplementary Figs. 3e and 4a). These results together suggest that the secretion signal peptide composed of the first 19 amino acids of ENDU-2 is necessary and sufficient to trigger secretion of a protein. Strikingly, we noticed that another transgenic reporter expressing 3xFlag::ENDU-2::EGFP was expressed strongly in the intestinal and also weakly in some head neurons, muscle cells in the head region, and anal depressor muscle cells, corroborating the report from Ujisawa et al. (Supplementary Fig. 3d). We speculate that the N-terminal 3xFlag fusion may prevent secretion by impairing binding of the secretion signal peptide by signal recognition particle (SRP), thus allowing detection of the weak expression in the muscles and neurons. In addition, expression in the head neurons and muscle cells might possibly be controlled by promoter signals localized in the first intron since this was the only sequence absent in the transgene expressing $\Delta_{ss}$endu-2::EGFP (Supplementary Fig. 3a), in which we failed to observe muscular or neuronal localization. Moreover, when we expressed *endu-2::EGFP* selectively either in the neurons (*unc-119* promoter), muscles (*myo-3* promoter) or intestine (*vha-6* promoter), ENDU-2::EGFP was always detected in the coelomocytes (Supplementary Fig. 4a), indicative for its secretion from these tissues. Notably, expressing *endu-2::EGFP* specifically in either neurons or muscles of heat-stressed animals resulted in ENDU-2::EGFP localization in the pharynx (Supplementary Fig. 4b), suggesting that either secretion or uptake of ENDU-2 in these tissues could be modulated by temperature.

**Secretion of ENDU-2 from the soma to the gonad protects germline immortality**. It is generally accepted that, in *C. elegans*, multi-copy extrachromosomal arrays are silenced and not expressed in the germline[12]. However, we found that extrachromosomal *endu-2::EGFP* transgenes rescued the Mrt germline phenotype of *endu-2(lf)* animals (Fig. 3a), suggesting that somatic ENDU-2 might preserve germline immortality across tissue boundaries. Our hypothesis was that secreted ENDU-2 could be endocytosed by the gonad. To test this assumption, we first asked whether ENDU-2 protein could be detected in the germline. All of our *endu-2::EGFP* transgenic reporters displayed weak expression levels unless secretion was blocked. Therefore, we could only occasionally observe faint ENDU-2::EGFP in few oocytes (Supplementary Fig. 4c). Via GFP-antibody staining we detected ENDU-2::EGFP in punctate structures in the germline at 15, 20, and 25 °C (Fig. 2c and Supplementary Fig. 5). In contrast, $\Delta_{ss}$ENDU-2::EGFP was not detected in the germline despite of its high expression level (Fig. 2c and Supplementary Fig. 3F). In addition, fusing the secretion signal peptide of another secreted protein SEL-1[13] to $\Delta_{ss}$ENDU-2::EGFP reactivated secretion and its localization in the germline (Fig. 2c and Supplementary Fig. 3e). These observations indicate that secretion of ENDU-2 is indispensable for its gonadal localization. Moreover, we performed single molecular FISH (smFISH) staining to visualize *endu-2* mRNA. *endu-2* mRNA signals were visible predominantly in the intestine, but were clearly absent from the germline (Fig. 2d). Furthermore, sequencing of RNA extracted from isolated gonads also failed to detect *endu-2* mRNA in the wild type gonad (Supplementary Data 3). Taken together, these results implicate that somatically expressed ENDU-2 protein is secreted and can be taken up by the gonad.

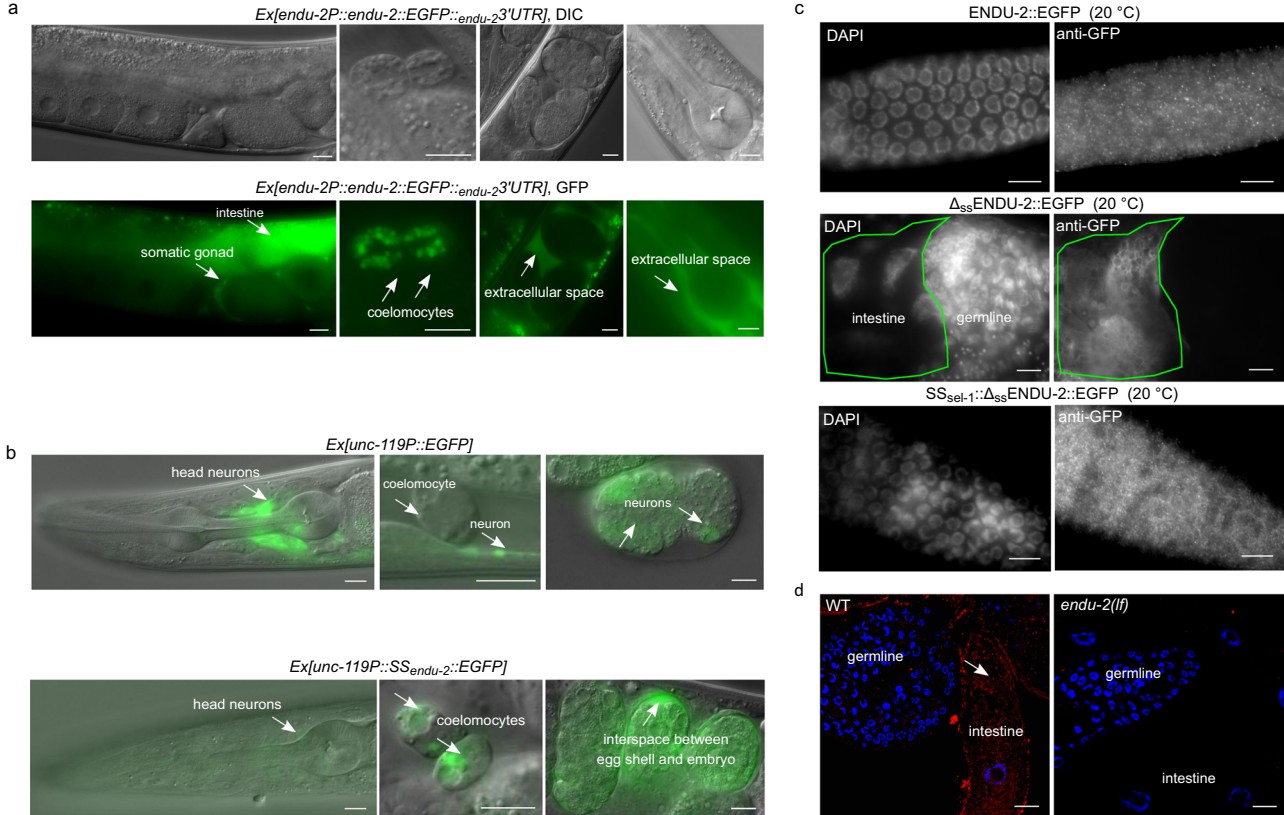

**Fig. 2 ENDU-2 is a secreted protein. a** Fluorescence micrographs of transgenic *endu-2(tm4977);byEx1814[endu-2P::endu-2::EGFP::_endu-2_3'UTR]* animals. ENDU-2::EGFP is detected in the intestine, the somatic gonad, coelomocytes, and extracellular space between the uterus and embryos. $N = 5$ independent experiments. **b** Fusion of the 1–19 amino acids of ENDU-2 (SS_endu-2_) to neuronal-specific expressed EGFP is sufficient for secretion of EGFP protein. Upper panels show localization of *unc-119P::EGFP* which is detected only in neuronal cells (strong) of animals and embryos. Lower panels are images showing localization of *unc-119P::SS_endu-2_::EGFP* which is detected in the neurons (weak), coelomocytes and interspace between egg shell and embryos, indicative of efficient secretion of EGFP mediated by the secretion signal peptide of ENDU-2. Images are representative for more than 20 animals, analyzed by using a ×40 objective. **c** Fluorescence micrographs of GFP-antibody staining of *endu-2(tm4977);byEx1375[endu-2P::endu-2::EGFP]*, *endu-2(tm4977);byEx1449[endu-2P::Δ_ss_endu-2::EGFP]* and *endu-2(tm4977);byEx1875[endu-2P::SS_sel-1_::Δ_ss_endu-2::EGFP]* transgenic animals at 20 °C. ENDU-2::EGFP ($n = 62$, 100%) and SS_sel-1_::Δ_ss_ENDU-2::EGFP ($n = 21$, 100%) but not Δ_ss_ENDU-2::EGFP ($n = 26$, 0%) is detected in the germline. $N = 2$ independent replicates. **d** smFISH staining reveals presence of *endu-2* mRNA in the intestine (white arrow), but not in the germline. Blue: DAPI stained nuclear DNA. Red: smFISH probes stained mRNA. $N = 4$. Scale bar 10 μm for all images in this Figure.

Next, we asked from which tissue ENDU-2 ensures germline immortality. Expressing ENDU-2 specifically in the intestine or the neurons, but not the muscle or somatic gonad, was sufficient to rescue the Mrt phenotype, indicating a non-cell-autonomous ENDU-2 signal from the neurons and intestine to the germline (Fig. 3b). In addition, expressing the secretion-deficient Δ_ss_ENDU-2::EGFP failed to rescue the Mrt phenotype (Fig. 3c). Furthermore, SS_sel-1_::Δ_ss_ENDU-2::EGFP rescued the Mrt phenotype (Fig. 3c), suggesting that guiding ENDU-2 into ER-Golgi secretory pathway via a canonical secretion signal suffices for functions of ENDU-2 in the germline.

To directly test whether loss of somatic *endu-2* expression is sufficient to induce a mortal germline, we performed *endu-2* RNAi knock-down in a *ppw-1* mutant background in which germline RNAi does not function[14]. Although neither wild type nor *ppw-1* mutants upon *endu-2* RNAi displayed a fully penetrant Mrt phenotype within 15 generation, the soma-specific *endu-2* RNAi resulted in gradually reduced brood sizes over generations in both *ppw-1* and wild type background, indicating that somatically expressed *endu-2* mRNA is required for normal reproduction (Supplementary Fig. 6a). The failure to obtain a fully penetrant Mrt phenotype in this experiment may be due to the relative inefficacy of RNAi in the nervous system, a tissue that expresses *endu-2* to ensure germline immortality.

**ENDU-2 affects the oocytes to preserve germline immortality independent of the nuclear RNAi pathway.** To know whether ENDU-2 affects oocytes or sperm to ensure germline immortality, we tested the parental contribution of ENDU-2 during sexual reproduction. For this purpose, we crossed *endu-2(+/+)* wild type parents with either males or hermaphrodites of *endu-2(−/−)* that had been grown at 25 °C for five to seven generations and displayed strongly reduced brood size and high percentage of sterility (Fig. 3d). Whereas the heterozygous progeny of *endu-2(+/+)* mothers had brood sizes with typically more than 100 F2 animals, the brood size of F1 cross-progeny derived from *endu-2(−/−)* mothers was as low as that of their mothers. We conclude that ENDU-2 function is required predominantly in the oocytes to preserve germline immortality.

The temperature-dependent Mrt phenotype of *endu-2* mutants resembles that of *hrde-1* mutants in the nuclear RNAi pathway[5,15]. In addition to loss of germline immortality, *hrde-1* is also defective in multigenerational inheritance of germline

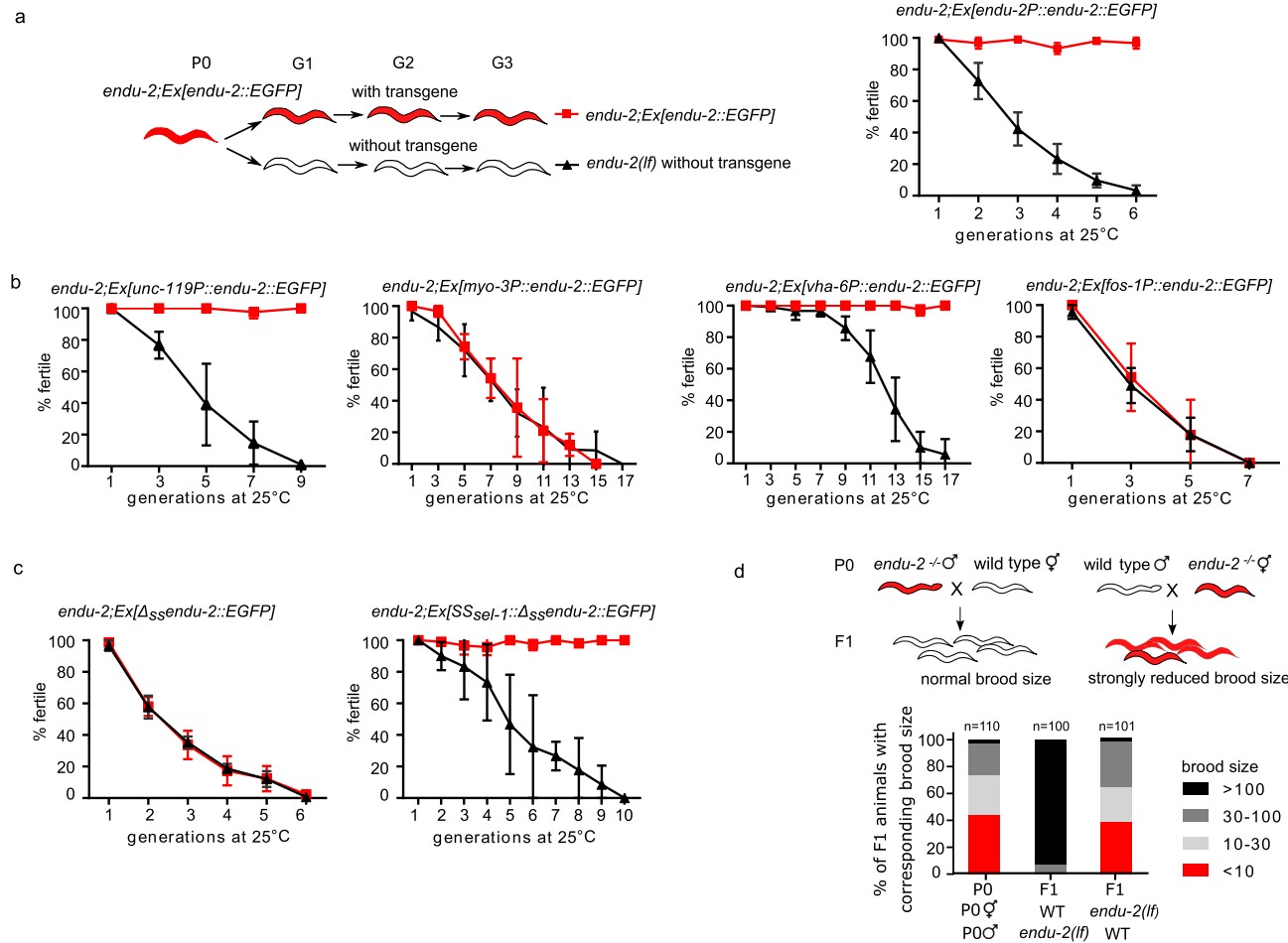

**Fig. 3 Maternally provided ENDU-2 from the soma contributes to maintenance of germline immortality. a** *endu-2(wt)::EGFP* rescues the Mrt phenotype at 25 °C. Left panel shows the experimental strategy for all rescue experiments with *endu-2* transgenes in this study. *endu-2(−/−)* daughter generation (G1) that has lost the extrachromosomal rescue constructs expressing *endu-2::EGFP* was isolated. This and the following *endu-2(−/−)* generations (G2 to Gn) were compared to *endu-2;Ex[endu-2::EGFP]* derived from the same P0 animal. Data are mean ± SD, N = 3. **b** Expression of *endu-2(wt)::EGFP* from neurons or intestine is sufficient to rescue the Mrt phenotype at 25 °C. Data are mean ± SD, N = 3. **c** SS_sel-1_::Δ_ss_endu-2::EGFP but not Δ_ss_endu-2::EGFP transgene rescues the Mrt phenotype at 25 °C. Data are mean ± SD, N = 3. **d** ENDU-2 primarily affects oocyte to maintain germline immortality. Data are pooled data from three biological replicates with similar tendencies in results.

RNAi. To test whether ENDU-2 acts in the nuclear RNAi pathway, we examined *oma-1* RNAi inheritance that suppresses embryonic lethality of an *oma-1* gain-of-function mutant for several generations[16]. Unlike the *hrde-1* mutants that lost inheritance of *oma-1* RNAi within two generations, effect of *oma-1* RNAi knock-down persisted for 5–6 generations both in *endu-2(lf)* and wild type animals (Supplementary Fig. 6b), suggesting that ENDU-2 does not act in the nuclear RNAi pathway to control transgenerational inheritance of germline RNAi. Therefore, maintenance of germline immortality by ENDU-2 probably functions via a mechanism distinct from that of the nuclear RNAi pathway.

**mRNA binding by ENDU-2, but not its mRNA-cleavage, is essential for maintaining germline immortality.** ENDU-2 harbors two XendoU domains, of which the C-terminally localized domain is more similar to both human EndoU and *Xenopus* XendoU (Supplementary Fig. 7). It was suggested recently that ENDU-2 might also function as RNA-binding protein[10]. To identify the RNAs bound by ENDU-2, we precipitated ENDU-2:: EGFP and analyzed co-immunoprecipitated RNAs by deep-sequencing (RIP-Seq). Since wild type ENDU-2 would potentially cleave its RNA targets, we thought that this might prevent

enrichment of intact RNA targets and their subsequent identification. Although we had no direct proof of an RNA cleaving activity of ENDU-2 yet, we reasoned that generation of an ENDU-2 variant that maintains RNA binding, but has lost RNA-cleavage activity, should facilitate RNA target detection. *Xenopus* XendoU mutants with E to Q exchanges at positions 161 or 167 of the EndoU domain lose RNA cleavage without reducing their RNA-binding activity[17]. The second glutamic acid (167E) is conserved in both XendoU domains of *C. elegans* ENDU-2 (175E and 460E), whereas the first (161E) is only found in the second XendoU domain (454E) (Supplementary Fig. 7b). We expressed both ENDU-2(E454Q)::EGFP and ENDU-2(E460Q)::EGFP variants in *endu-2(lf)* background and performed RIP-Seq with these two strains in additional to wild type ENDU-2::EGFP. By plotting normalized reads (RPM) of each transcript we were able to investigate RNA-binding affinity of ENDU-2 under different conditions. In general, all three ENDU-2 variants tested showed stronger RNA-binding affinity at 15 °C than at 25 °C (Fig. 4a). E460Q showed weaker RNA-binding already at 15 °C than wild type ENDU-2 and almost completely lost RNA-binding capacity at 25 °C. In addition, ENDU-2(E454Q) displayed stronger RNA-binding activity than ENDU-2(wt) at 15 °C. Therefore, we used RIP-Seq data of ENDU-2(E454Q) at 15 °C for detecting RNAs

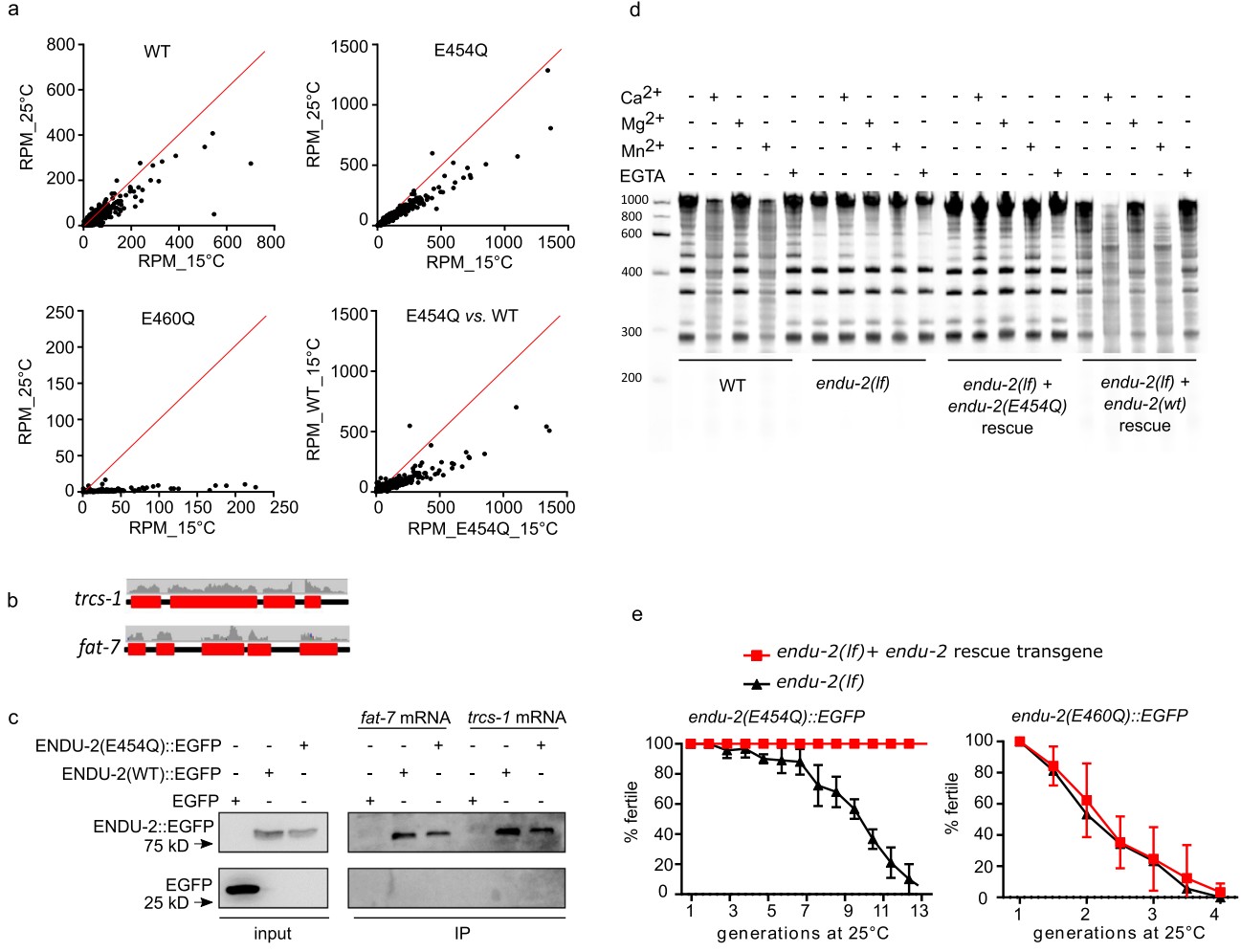

**Fig. 4 RNA binding but not RNA-cleavage activity of ENDU-2 protects germline immortality. a** Comparison of RNA binding activities of ENDU-2(wt):: EGFP, ENDU-2(E454Q)::EGFP, and ENDU-2(E460Q)::EGFP at 15 and 25 °C. Shown are plotted normalized reads (RPM) from RIP-Seq of each identified transcript under different conditions. **b** Mapping of the RIP-Seq reads of two representative binding targets of ENDU-2. Only fragments of mRNA exons (red boxes) but not introns (black lines) were co-immunoprecipitated with ENDU-2. **c** ENDU-2::EGFP variants bind to selected mRNAs in vitro. Shown are western blots to detect proteins binding to *fat-7* and *trcs-1* mRNA, respectively. EGFP is a negative control. $N = 3$ biological replicates. The uncropped blots are included in Supplementary Fig. 11b. **d** Wild type ENDU-2, but not ENDU-2(E454Q), leads to RNA decay (smear) in a $Ca^{2+}$ and $Mn^{2+}$ dependent manner, $N = 3$. Shown is fused images of two gels from one experiment. The results of additional two biological replicates are shown in Supplementary Fig. 8a. **e** *endu-2(E454Q)::EGFP* but not *endu-2(E460Q)::EGFP* transgene rescues the Mrt phenotype. Data are mean ± SD, $N = 3$. Both *endu-2(tm4977)* and *endu-2(tm4977)* carrying *endu-2(E454Q)::EGFP* or *endu-2(E460Q)::EGFP* transgenes were decedents of one single P0 animal carrying the respective transgenes.

bound by ENDU-2. 5920 transcripts were co-immunoprecipitated with ENDU-2(E454Q) (Supplementary Data 1). Most of them (>99%) were protein-coding transcripts, the remaining were non-coding RNAs (5 snoRNAs, 27 pseudogenes, 5 ncRNAs, and 1 lincRNA). In addition, reads distribution analysis showed >99% of sequenced reads were mapped to exons, suggesting that ENDU-2 (E454Q) primarily binds to processed mRNAs (Fig. 4b). We performed in vitro mRNA-binding assays with recombinant ENDU-2 proteins and two selected mRNA targets from the RIP-Seq data and confirmed that both ENDU-2 and ENDU-2(E454Q) directly interact with these mRNAs (Fig. 4c and Supplementary Fig. 11b).

Studies of *Xenopus* XendoU had shown that that the RNA hydrolysis activity of XendoU requires $Mn^{2+}$ or $Ca^{2+}$ [18]. ENDU-2 seems to require similar conditions, since the addition of 5 mM $Mn^{2+}$ or $Ca^{2+}$ in the buffer medium, but not 5 mM $Mg^{2+}$, led to degradation of bulk RNA in the wild type worm lysates (Fig. 4d and Supplementary Fig. 8). Bulk RNA degradation was strongly

reduced in *endu-2(lf)* animals. In addition, expression of the wild type *endu-2::EGFP* transgene, but not of *endu-2(E454Q)::EGFP*, restored RNA decay in *endu-2(lf)* mutants, supporting our hypothesis that ENDU-2(E454Q) lost RNA-cleavage capacity despite its increased RNA-binding affinity. Surprisingly, extra-chromosomal expression of ENDU-2(E454Q), but not of ENDU-2(E460Q), fully rescued the Mrt phenotype of *endu-2(lf)* at 25 °C (Fig. 4e). We conclude that RNA binding, rather than RNA-cleavage activity of ENDU-2, is essential in the germline to maintain stem cell immortality at elevated temperature.

**ENDU-2 negatively regulates somatic mRNA abundance via its endoribonuclease activity.** To test if ENDU-2 affects mRNA levels, we performed microarray experiments at 25 °C to determine alterations in the transcriptome mediated by ENDU-2 (procedure of sample preparation is illustrated in Supplementary Fig. 9a). A comparison of differential gene expression in

*endu-2*(−) and *endu-2*(+) backgrounds suggested a similar number of transcripts being up- and downregulated by *endu-2* (fold-change >2) (Fig. 5a). Among the 258 transcripts downregulated in *endu-2(lf)* animals, germline expressed genes (*n* = 206) were over-represented (Supplementary Data 2). qPCR quantifications of two selected genes *cav-1* and *trsc-1* confirmed our microarray results and revealed that their expression levels strongly decreased in *endu-2* mutants compared to wild type animals at 25 °C (Supplementary Fig. 10c and 10d). However, neither a germline expressed *cav-1::GFP* reporter nor smFISH

staining of *trcs-1* confirmed significantly altered germline expression in situ (Supplementary Fig. 10a, 10b and 10d). As the gonads of *endu-2(lf)* mutants at 25 °C became significantly smaller (Supplementary Fig. 9b), we speculate that apparent downregulation of germline mRNA within the transcriptome data set may be caused by reduced sample size of the gonad in *endu-2* mutants.

Therefore, we put our focus on genes that were upregulated in *endu-2(lf)* background. Thirty-two percent (76 out of 237) of these transcripts were candidates for direct ENDU-2 targets, since

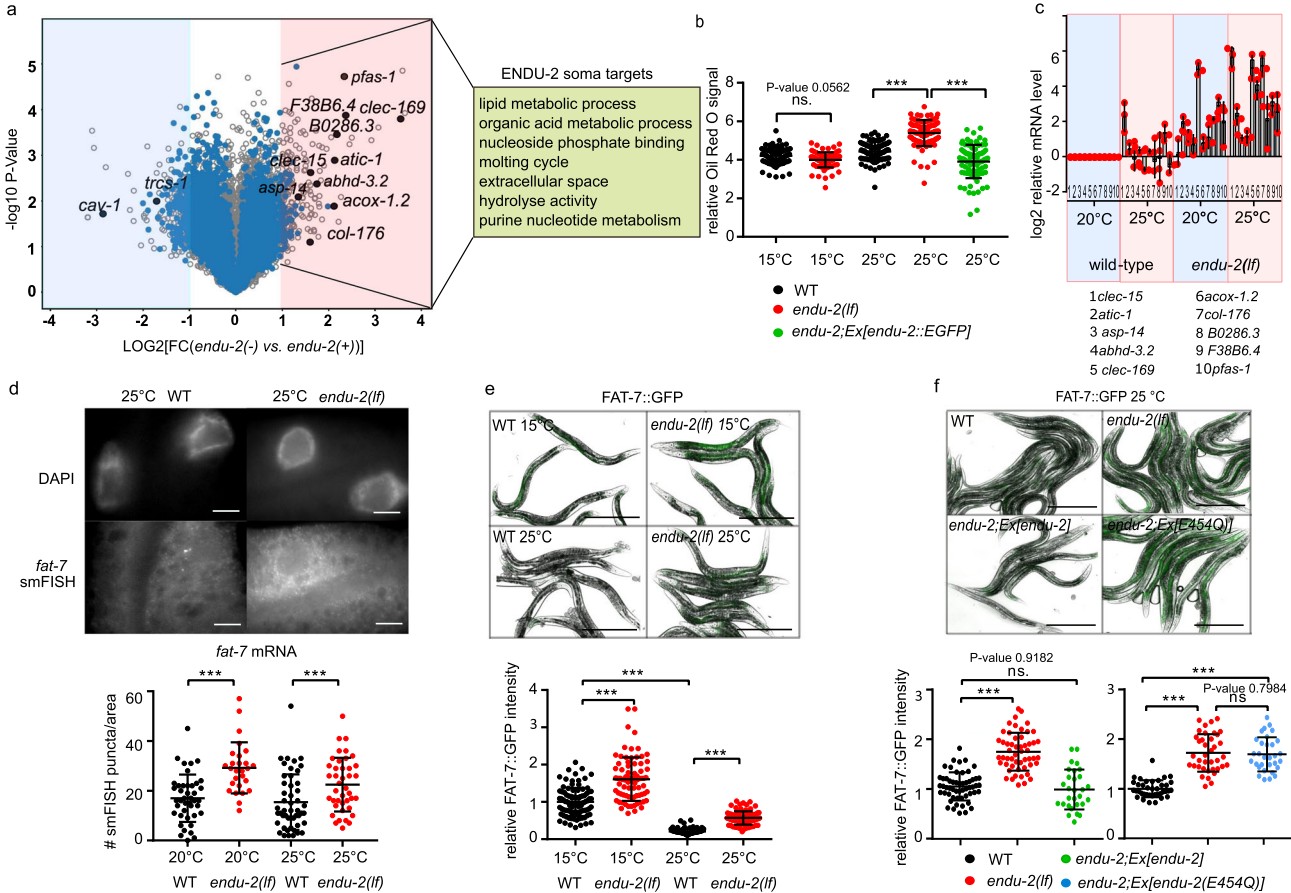

**Fig. 5 ENDU-2 negatively regulates mRNA abundance in the soma. a** Volcano blot of microarray results comparing *endu-2(tm4977)* mutant and *endu-2 (tm4977)* animals carrying the rescuing *endu-2::EGFP* transgene at 25 °C. *N* = 3 biological replicates. The dots in the bright red or blue areas are differentially regulated transcripts (FC > 2 with two-tailed *P*-value <0.05). The blue dots mark transcripts bound by ENDU-2. Transcripts with blue and black label were ENDU-2 associated according to the RIP-Seq result and those with black labels were verified with qPCR. **b** ENDU-2 negatively regulates lipid content at 25 °C. Shown are mean ± SD of quantification of relative Oil Red O stained signal in wild type, *endu-2(tm4977)* as well as *endu-2(tm4977); byEx1375[endu-2::EGFP]* day-one adult animals. Numbers of examined samples over three independent experiments: n = 80 for wild type and *endu-2 (tm4977)* at 15 °C, wild type at 25 °C, n = 89 for *endu-2(tm4977)* at 25 °C, n = 86 for *endu-2(tm4977);byEx1375[endu-2::EGFP]* at 25 °C. Statistical test with one-way ANOVA. *P*-values were calculated with Tukey's multiple comparison test. *P*-value for WT vs. *endu-2(tm4977)* at 15 °C: 0.0562. ***P*-value <0.0001. **c** qPCR to quantitate selected ENDU-2 target mRNAs in the soma at 20 and 25 °C. Data are mean ± SD, *N* = three biological replicates. **d** Fluorescent images of smFISH staining of the ENDU-2 target *fat-7* mRNA and quantification of the smFISH stained mRNA foci per examined area in wild type and *endu-2(tm4977)* animals. Scale bar 10 μm. Data are mean ± SD. Numbers of examined samples over three independent experiments: wild type at 20 °C n = 42, *endu-2(tm4977)* at 20 °C n = 31, wild type at 25 °C n = 49, *endu-2(tm4977)* at 25 °C n = 41. Statistical test with one-way ANOVA. *P*-values were calculated with Tukey's multiple comparison test. ***P*-value <0.0001. **e** Fluorescent images and quantification of fluorescence intensity of FAT-7:: GFP in wild type and *endu-2(tm4977)* day-one adult animals at different temperatures. Data are mean ± SD. Numbers of examined samples over three independent experiments: wild type at 15 °C n = 92, *endu-2(tm4977)* at 15 °C n = 80, wild type at 25 °C n = 102, *endu-2(tm4977)* at 25 °C n = 89. Statistical test with one-way ANOVA. *P*-values were calculated with Tukey's multiple comparison test. ***P*-value <0.0001. Scale bar 500 μm.
**f** Fluorescent images and quantification of fluorescence intensity of FAT-7::GFP in wild type, *endu-2(tm4977)*, *endu-2(tm4977);Ex[endu-2(wt)]* and *endu-2 (tm4977);Ex[endu-2(E454Q)]* day-one adult animals at 25 °C. Data are mean ± SD. Numbers of examined samples over three independent experiments: For *endu-2(wt)* rescue experiment, wild type n = 55, *endu-2(tm4977)* n = 53, *endu-2(tm4977);Ex[endu-2(wt)]* n = 29. For *endu-2(E454Q)* rescue experiment, wild type n = 38, *endu-2(tm4977)* n = 38, *endu-2(tm4977);Ex[endu-2(E454Q)]* n = 31. Statistical test with one-way ANOVA. *P*-values were calculated with Tukey's multiple comparison test. *P*-value for WT vs. *endu-2(tm4977);Ex[endu-2(wt)]*: 0.9182, for *endu-2(tm4977)* vs. *endu-2(tm4977);Ex[endu-2(E454Q)]*: 0.7984. ***P*-value <0.0001. Scale bar 500 μm.

they had been co-immunoprecipitated with ENDU-2 (Supplementary Data 1). In addition, the vast majority of them (62 out of 76) did not have increased expression in the gonad of *endu-2(lf)* animals (see next chapter and Supplementary Data 1). This suggests that, at minimum, these 62 transcripts are down-regulated by ENDU-2 in the somatic tissues. GO term analyses implicated these somatic ENDU-2 targets in regulation of various metabolic processes (Fig. 5a). Consistently, we found that loss of *endu-2* resulted in increased lipid content at 25 °C but not at 15 °C (Fig. 5b). Taken together, these results indicate a putative regulatory role of ENDU-2 in limiting abundance of the mRNAs that are primarily involved in metabolic functions in the soma.

As some *endu-2(lf)* phenotypes were temperature-dependent, we asked whether ENDU-2 regulates mRNA abundance in response to temperature alterations. To test this, we performed qPCR to quantitate ten selected mRNAs that were bound and downregulated by ENDU-2. We found that mRNA levels of these targets were upregulated in *endu-2* mutant background, compared to wild type, already at standard growth temperature (20 °C), and were even more abundant at 25 °C (Fig. 5c). Increased temperature, on the other hand, did not strongly affect the abundance of most of these mRNAs in wild type background. These results suggest a temperature-dependent negative influence of ENDU-2 on the levels of these mRNA targets. We also performed smFISH to inspect the influence of ENDU-2 on the mRNA level of yet another target *fat-7*. *fat-7* mRNA was only detected in the intestine and *endu-2(−)* animals had higher *fat-7* transcript levels than wild type at both 20 and 25 °C (Fig. 5d). Moreover, we used a *fat-7::GFP* translational fusion reporter to monitor FAT-7::GFP protein level. Wild type animals showed reduced FAT-7::GFP levels at 25 °C vs. 15 °C (Fig. 5e). *endu-2(lf)* displayed stronger FAT-7::GFP expression levels at both temperatures. Furthermore, transgenic expression of *endu-2(wt)* but not the *endu-2(E454Q)* transgene (that has lost RNA-cleavage activity) restored decreased FAT-7::GFP level (Fig. 5f). We conclude that ENDU-2 mediated mRNA-cleavage is required for decreasing expression of at least some of its somatic target genes, such as *fat-7*.

**ENDU-2 prevents misexpression of soma-specific genes in the germline**. To investigate germline transcriptomes regulated by ENDU-2, we isolated gonads from wild type and *endu-2(lf)* animals which had been grown at 25 °C from L1 stage, and sequenced two biological replicates of polyadenylated RNA from each strain. A gene with normalized reads of RPKM > 1 was scored as being expressed. We identified 8356 expressed genes in the gonad of wild type animals at 25 °C, among which 88% (7365) have been reported as expressed in the wild type gonad at 20 °C[19]. The following DESeq2 analysis showed that upregulation of mRNA levels was the prevalent change in the gonad of *endu-2* mutants (422 upregulated, 35 downregulated, FC > 2, *P*-value <0.05; Fig. 6a, Supplementary Data 3). Only genes upregulated in *endu-2(lf)* background were enriched in ENDU-2 binding, as 218 out of 422 upregulated genes (*P*-value < 0.0001, Chi-square test) and 11 out of 35 downregulated genes (*P*-value = 0.7675, Chi-square test) were co-immunoprecipitated with ENDU-2. We considered them as direct targets of ENDU-2 and grouped them in two classes, depending on whether they are up- (class I) or downregulated (class II) in *endu-2(lf)* background (Fig. 6b). Expression of a class I targets is repressed, while class II genes are activated by wild type ENDU-2 activity. Consistently, we noticed that most of the class I, but not the class II target genes, were not, or very lowly, expressed in the gonad of wild type animals (Fig. 6c). Additional tissue enrichment analysis suggested that the class I targets are enriched in genes expressed in the somatic

tissues (intestine, neurons, muscles, somatic gonad, and pharynx). GO term enrichment analysis revealed that the major function of class I targets are involved in immune and defense responses (Fig. 6b). We wondered whether the class I target genes were preferentially soma-specific genes that are misexpressed in the germline in the absence of ENDU-2. To validate this assumption, we combined our wild type gonad transcriptome data with a previous published whole animal's transcriptome analysis at 25 °C[20] and calculated a soma enrichment factor (SEF) for each transcript to estimate the expression ratio between the soma and the gonad. Class I, but not class II, targets had on average significantly higher SEF values than the total mRNAs (Fig. 6d), suggesting that the class I genes are predominantly expressed in the soma, but only expressed at low levels in the germline of wild type animals. As maintenance of germline immortality by ENDU-2 does not require its RNA-cleavage activity, we additionally sequenced mRNA from isolated gonads of *endu-2(lf)* mutant carrying the *endu-2(E454Q)::EGFP* transgene. Expressing *endu-2 (E454Q)::EGFP* was sufficient to repress germline expression of most class I targets (Fig. 6e and f). In summary, our data indicate that the inhibitory role of ENDU-2 to prevent germline expression of soma-specific genes is probably independent of its RNA-cleavage activity.

**Discussion**

The task of the germline is to maintain pluripotency and immortality for an accurate transmission of genetic and epigenetic information between generations. Despite emerging studies about molecular mechanisms protecting the germline from the impacts of stress, one obvious open question is whether the germline responds directly to environmental signals, such as nutrients or stress, or rather reacts to alterations in somatic signaling. A direct stress response seems to be less advantageous from an economical point of view, since this requires, in a pluripotent stem cell, the expression of an entire hierarchy of mechanisms, involving the sensation of stress, signaling, as well as responsive pathways to enable a dynamic and rapid response to a wide range of environmental cues. In contrast, the second alternative would be more beneficial, as differentiated somatic cells are specialized to sense environmental changes. In such a case, molecular messengers, such as secreted hormones and ligands, are required to shuttle between somatic and germline tissues to adjust a proper response in the germline. Previous studies have revealed that microRNAs could act as such messengers mediating communication between soma and germline[21,22]. Here, we show that secretion of a conserved endoribonuclease ENDU-2 from the soma prevents misexpression of soma-specific gene in the germline and preserve germline immortality at elevated temperature. This finding, together with another recent study reporting regulation of germline proliferation by intestinal ENDU-2 in response to thymidine imbalance[11], strongly suggests ENDU-2 as a crucial molecule mediating non-cell-autonomous stress responses.

Our data implicate ENDU-2 in enabling long-range communication between cells, mediated via its secretion signal peptide (Fig. 2b). Whereas an earlier study had claimed neurons and muscles as the tissues primarily expressing *endu-2*[10], our data, consistent with the report from Jia et al., suggest intestine as the most important organ to produce ENDU-2, whereas expression in neuronal, muscular, and somatic gonad tissues is rather weak (Fig. 2a and Supplementary Fig. 3). We compared, for this analysis, both RNA and protein levels of ENDU-2 as well as different ENDU-2 variants with truncations of the N-terminal secretion signal peptide ($\Delta_{ss}$ENDU-2 and SS$_{sel-1}$::$\Delta_{ss}$ENDU-2). We could demonstrate that the tissues containing ENDU-2 protein differ

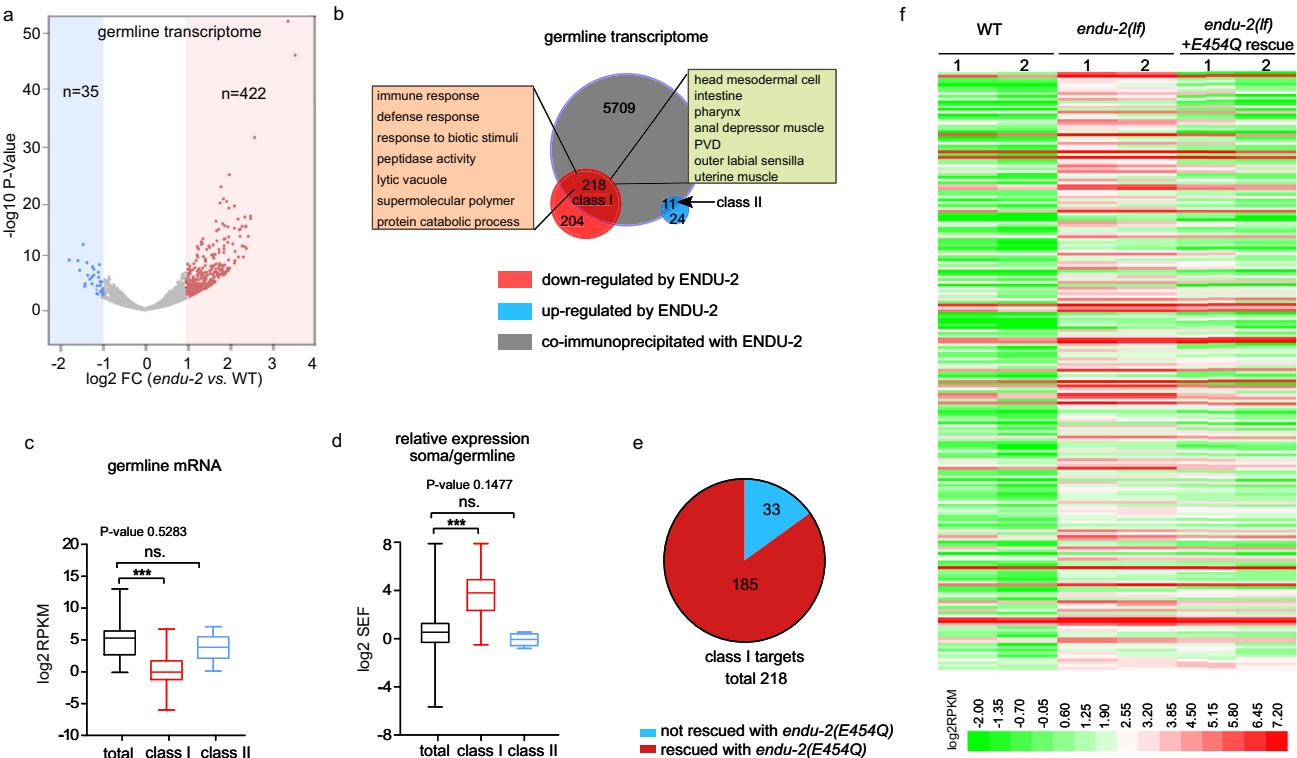

**Fig. 6 ENDU-2 prevents misexpression of soma-specific genes in the germline. a** Volcano plot of gonadal expressed genes in *endu-2(tm4977)* relative to wild type day-one adult animals grown up at 25 °C. Two biological replicates were obtained for each strain. Significantly misregulated genes (FC > 2, FDR < 0.05) were calculated with DESeq2 and are shown in red or blue. *n* refers to the total numbers of up- or downregulated genes. **b** Comparison of the total transcripts co-immunoprecipitated with ENDU-2 and ENDU-2 regulated genes in the gonad. **c** Comparison of expression levels (RPKM) of total gonad-expressed genes (*n* = 8356), ENDU-2 class I (*n* = 218) and class II targets (*n* = 11) in wild type animals. Shown are boxes extending from the 25th to the 75th percentile, with the median indicated by the horizontal line and Min/Max whiskers. Statistical test with one-way ANOVA. *P*-values were calculated with Dunnett's multiple comparison test. *P*-value for total vs. class II targets: 0.5283. ***P*-value <0.0001. **d** Comparison of relative expression between the soma and the gonad (SEF) of the total transcripts (*n* = 12371), ENDU-2 class I (*n* = 218), and class II (*n* = 11) targets in wild type animals. Shown are boxes extending from the 25th to the 75th percentile, with the median indicated by the horizontal line and Min/Max whiskers. Statistical test with one-way ANOVA. *P*-values were calculated with Dunnett's multiple comparison test. *P*-value for total vs. class II targets: 0.1477. ***P*-value <0.0001. **e** The majority of the alerted expression of the germline class I targets is rescued by *endu-2(E454Q)::EGFP* transgene. **f** Heat map of ENDU-2 repressed germline genes in wild type, *endu-2(tm4977)* and *endu-2;Ex[endu-2(E454Q)::EGFP]* animals.

from those that harbor *endu-2* mRNA and presence of a secretion signal peptide is necessary and sufficient to target ENDU-2 to the secretory pathway to reach distant cells.

Our data suggest that ENDU-2 production and function may occur in different tissues, and that secretion of ENDU-2 allows the control of mRNA abundance in the distance. A simple assumption would be that ENDU-2 has similar functions in any of its target tissues, no matter from which cells it is expressed. However, only neuronally or intestinally expressed ENDU-2 was able to rescue germline immortality (Fig. 3c), indicating existence of functional difference of ENDU-2 from distinct origins. Tissue-specific interactors or modifiers of ENDU-2 are probably crucial to specify its individual activities. It is currently not known whether there are isoform-specific functions of ENDU-2, or whether ENDU-2 protein activity is modulated by protein modifications or interactions.

We distinguished two activities of ENDU-2: mRNA binding and mRNA-cleavage. RIP-Seq and transcriptomic results demonstrated that the levels of only about 10% of the mRNAs to which ENDU-2 binds changed in *endu-2(lf)* mutants at elevated temperature (Figs. 4–6), suggesting that ENDU-2 is able to discriminate whether its targets are only bound or are hydrolyzed. Such a scenario has also been proposed for SMG-2/UPF1, the core factor of the non-sense mediated decay (NMD) complex that

degrades only a small fraction of the RNAs it binds to[23]. In addition, binding to mRNA might generate a platform for interaction with other RNA-binding proteins which may act together with ENDU-2 to regulate RNA stability. ENDU-2 represses expression of its somatic target genes probably via direct RNA degradation, while it utilizes RNA binding, but not cleavage activity, to prevent misexpression of soma-specific genes in the germline, suggesting rather an indirect mechanism of ENDU-2 in the germline to prevent a gene expression program specific for differentiated somatic cells. Whether this is regulated at transcriptional or post-transcriptional level is currently unknown. Another puzzle is how germline localized ENDU-2 selectively represses expression of soma-specific genes. Germ granules of *C. elegans* have been shown to inhibit misexpression of somatic transcripts[24]. The punctate localization pattern of ENDU-2 in the germline indicates its possible localization in the germ granules (Fig. 2c). Whether ENDU-2 affects functions of the germ granules to restrict somatic transcripts should be studied in the future. We speculate that association of ENDU-2 with distinct protein complexes in the soma and germline may differentiate its functionalities and adjust gene expression in the soma and germline in response to temperature alterations.

*Xenopus* EndoU was suggested to both control snoRNA biogenesis in the nucleolus and cleave mRNAs in the cytosol[17,18].

Mouse EndoU has been reported to regulate peripheral B cell survival via reducing c-Myc mRNA level[25]. Inactivation of neuronal Drosophila DendoU causes neurotoxicity partially by downregulation of dTDP-43[26]. SARS and SARS-Covid-2 EndoU homologs Nps15/NendoU have been implicated in virus replication as well as limitation of host innate immune response, although the detailed functions are still enigmatic[27]. Our work here suggests that *C. elegans* ENDU-2, in response to environmental stimuli such as temperature elevation, can exert its function in tissues that different from those expressing it. Therefore, ENDU-2 is a previously undescribed candidate for transmitting environmental signals across tissue boundaries, most notably involving signaling to the germline. A secretion signal peptide is also present in the human EndoU homolog that is expressed in the placenta and detected in the serum[28]. Therefore, the potential conservation of its non-cell-autonomous functions could open a research area to study intercellular communication. EndoU, thus, could be involved in a mechanism by which environmental influences and experiences of the somatic tissues are transmitted into the reproductive system.

## Methods

**Strains.** The *C. elegans* N2 (Bristol) strain was used as wild type in all experiments in this study. Mutant strains: KHR83 *endu-2(tm4977)*, outcrossed 8x with our N2. BR7130 *endu-2(by188)*, NL3511 *ppw-1(pk1425)*, TX20 *oma-1(zu405)*, BR7510 *oma-1(zu405);endu-2(tm4977)*, BR8649 *oma-1(zu405);hrde-1(tm1200)*, BR7205 *endu-2 (by190[endu-2::EGFP])*, BR7089 *byEx1315[endu-2P::EGFP;rol-6(su1006)]*, BR8657 *endu-2(tm4977);byEx1814[endu-2P::endu-2::EGFP::_endu-2_3'UTR;rol-6(su1006);]*, BR7295 *endu-2(tm4977);byEx1375[endu-2P::endu-2::EGFP;myo-2P::mCherry]*, BR5402 *byEx749[unc-119P::EGFP;rol-6(su1006)]*, BR8672 *byEx1821[unc-119P::SS_endu-2_::EGFP;rol-6(su1006)]*, BR7827 *endu-2(tm4977);byEx1551[vha-6P::endu-2::EGFP;3xFlag;myo-2P::mCherry]*, BR7332 *endu-2(tm4977);byEx1379[fos-1P::endu-2::EGFP;myo-2P::mCherry]*, BR8551 *endu-2(tm4977);byEx1795[unc-119P::endu-2::EGFP;rol-6(su1006)]*, BR8662 *endu-2(tm4977);byEx1816[myo-3P::endu-2::EGFP;3xFlag;myo-2P::mCherry]*, BR7512 *endu-2(tm4977);byEx1449[endu-2P::Δ_ss_endu-2::EGFP;rol-6(su1006);myo-2P::mCherry]*, BR8747 *endu-2(tm4977);byEx1847[endu-2P::Δ_ss_endu-2::EGFP::_endu-2_3'UTR;myo-2P::mCherry]*, BR8821 *endu-2(tm4977);byEx1875[endu-2P::SS_sel-1_::Δ_ss_endu-2::EGFP; myo-2P::mCherry]*, BR7680 *endu-2(tm4977);byEx1492[endu-2P::endu-2(E454Q)::EGFP::3xFlag;myo-2P::mCherry]*, BR7683 *endu-2(tm4977);byEx1495[endu-2P::endu-2(E460Q)::EGFP::3xFLAG;myo-2P::mCherry]*, DMS303 *nIs590[fat-7p::fat-7::GFP + lin15(+)]*, BR8317 *endu-2(tm4977);nIs590[fat-7p::fat-7::GFP + lin-15(+)]*, BR8754 *endu-2(tm4977);nIs590[fat-7p::fat-7::GFP + lin-15(+)];byEx1853[endu-2P::endu-2(E454Q);myo-2P::mCherry]*, BR8775 *endu-2(tm4977);nIs590[fat-7p::fat-7::GFP + lin-15(+)];byEx1860 [endu-2P::endu-2(E460Q)::EGFP;myo-2P::mCherry]*, RT688 *unc-119(ed3);pwIs28[pie-1p::cav-1::GFP(7)+unc-119(+)]*, BR8358 *endu-2(tm4977);pwIs28[pie-1p::cav-1::GFP(7)+unc-119(+)]*. For RIP-Seq the following strains were used: BR7802 *endu-2(tm4977);byIs240[endu-2P::endu-2(E460Q)::EGFP::3xFLAG;myo-2p::mCherry]*, BR7803 *endu-2(tm4977);byIs241[endu-2P::endu-2(E454Q)::EGFP::3xFLAG;myo-2P::mCherry]*, BR7205 *endu-2(by190[endu-2::EGFP])* and *Ex[ife-2P::GFP]* (as IP control, gift from Tavernarakis Lab). For the RNA-cleavage assay, BR8311 *endu-2(tm4977);byIs267 [endu-2P::endu-2::EGFP;myo-2::mCherry]* and BR7803 *endu-2(tm4977);byIs241 [endu-2P::endu-2(E454Q)::EGFP;3xFlag;myo-2P::mCherry]* were used in additional to N2 and *endu-2(tm4977)* strains. *endu-2(tm4977)* allele was used for all experiments, if not noted otherwise. Except for the Fig. 1a and Supplementary Fig. 2, all the *endu-2(lf)* mutants were the granddaughter generation (G2) from an *endu-2 (tm4977)* carrying transgenic *endu-2* rescue strains whose parents (G1) had lost the rescue transgene. Information for extrachromosomal transgenic strains generated in this study is summarized in the Supplementary Data 5.

**Plasmids.** To construct an EGFP transcriptional fusion reporter of *endu-2* (pBY3798), a 4691 bp genomic fragment upstream of the *endu-2* ATG was PCR amplified and inserted into pEGFP-N1 with Eco47III/BglII sites. For translational fusion reporters, 5487 bp *endu-2* genomic region was cloned into pBY3798 with BglII/SmaI sites to receive *endu-2P::endu-2::EGFP* (pBY3800), the *endu-2* 3'UTR was inserted into pBY3800 at NotI site to receive *endu-2P::endu-2::EGFP::_endu-2_3' UTR* (pBY4137). *Δ_ss_endu-2::EGFP* (pBY3843) and *Δ_ss_endu-2::EGFP::_endu-2_3'UTR* construct (pBY4172) was generated by removing sequences encoding the N-terminal 2–19 amino acids from pBY3800 and pBY4137, respectively. The *endu-2P::SS_sel-1_::Δ_ss_endu-2::EGFP* construct (pBY4194) was generated by inserting the sequence encoding the amino acids 1–20 of *sel-1* into the BglII site of pBY3843. A *endu-2P::3xFlag::endu-2::EGFP::_endu-2_3'UTR* expressing construct (pBY4138) was generated by inserting 3xFlag encoding sequence and *endu-2* genomic region into pBY3798 with BglII and BglII/SmaI sites, respectively. The 3'UTR of *endu-2* was inserted via Gibson ligation. The *unc-119P::EGFP* expressing construct (pBY2941)

was made by fusing the 2.1 kb *unc-119* promoter region into pEGFP-N1 with Eco47III/BglII sites. The *unc-119P::SS_endu-2_::EGFP* construct pBY4148 was generated by inserting the first 57 nucleotides of *endu-2* cDNA into pBY2941 with BglII/AgeI sites. For expressing recombinant ENDU-2::EGFP in HEK 293T cells, wild type *endu-2* cDNA was cloned into a 3xFlag containing pEGFP-N1 with XhoI/SmaI (pBY3878). Site-directed mutagenesis was performed to receive *endu-2 (E454Q)::EGFP::3xFlag* (pBY3894) and *endu-2(E460Q)::EGFP::3xFlag* mutants (pBY3895). The fragments *endu-2::EGFP::3xFlag* from pBY3878, *endu-2(E454Q):: EGFP::3xFlag* from pBY3894 and *endu-2(E460Q)::EGFP::3xFlag* from pBY3895 were inserted into pBY3798 to receive constructs for generating *endu-2P::endu-2:: EGFP::3xFlag* (pBY3892), *endu-2P::endu-2(E454Q)::EGFP::3xFlag* (pBY3897) and *endu-2P::endu-2(E460Q)::EGFP::3xFlag* (pBY3898) reporter strains. *EGFP::3xFlag* fragments in pBY3892 and pBY3897 were removed to receive pBY4188 *endu-2P:: endu-2* and pBY4066 *endu-2P::endu-2(E454Q)*. Somatic gonad specific expression was achieved via PCR amplifying and insertion of *endu-2* cDNA into a *fos-1P::GFP* plasmid (a gift from David Sherwood lab) with SalI and SmaI sites (pBY3833). For intestinal specific expression, a 1.2 kb *vha-6* promoter was inserted at Eco47III/XhoI sites into pBY3878 to receive *vha-6P::endu-2::EGFP::3xFlag* (pBY3937). A 2.1 kb *unc-119* promoter was inserted into pBY3878 at Eco47III/BglII sites to receive *unc-119P::endu-2(cDNA)::EGFP::3xFlag* (pBY4127) for neuronal expression. A 2.4 kb *myo-3* promoter region was cloned into pBY3878 to receive *myo-3P::endu-2:: EGFP::3xFlag* (pBY4135) for muscular specific *endu-2* expression.

**Scoring of Mrt phenotype.** Mrt phenotype was examined by quantification of brood size by separating 15 L4 animals on single plates and counting number of progeny per animal, or transferring six L4 larvae onto new agar plates and scoring percentage of the fertile animals 36 h after the L4 stage in the next generation. Both *endu-2(tm4977)* and *endu-2(by188)* were further outcrossed 4× with N2 before scoring Mrt phenotype for Fig. 1a and Supplementary Fig. 2a. For brood size recovery at 15 °C, wild type and outcrossed *endu-2(tm1977)* animals were maintained at 25 °C for 6 generations until *endu-2(tm4977)* showed strong sterile phenotype. Fifteen L1-L2 animals from the 6th generation were shifted to 15 °C and designated as G1. For Mrt rescue experiment, *endu-2(tm4977)* progenies of *endu-2(tm4977);byEx1375[endu-2P::endu-2::EGFP;myo-2P::mCherry]*, *endu-2(tm4977); byEx1449[endu-2P::Δ_ss_endu-2::EGFP;myo-2P::mCherry]*, *endu-2(tm4977);byEx1875 [endu-2P::SS_sel-1_::Δ_ss_endu-2::EGFP;myo-2P::mCherry]*, *endu-2(tm4977);byEx1488 [endu-2P::endu-2(E454Q)::EGFP::3xFLAG;myo-2P::mCherry]* and *endu-2(tm4977); byEx1488[endu-2P::endu-2(E460Q)::EGFP::3xFLAG;myo-2P::mCherry]* animals (P0) were isolated and designated as G1.

**Germline proliferation.** The 'number of nuclei in the proliferative zone' included all the germ nuclei between the distal tip and the transition zone of day-one adult animals (24 h after mid-L4 larval stage). For visualizing and counting germ cell nuclei in the proliferating zone of the gonad, dissected gonads were fixed in methanol and suspended in PBST containing 0.1% Tween 20 and 2 μg/ml 4′,6-diamidino-2-phenylindole (DAPI) before microscopy. Z-stack images of animals were collected and the numbers of the germ cells were counted with a publicly available ImageJ Cell Counter plug-in originally written by Kurt De Vos at the University of Sheffield, Academic Neurology.

**Lifespan.** Lifespan at 20 °C was initiated at L4 stage. As *endu-2(tm4977)* animals display strong egg-laying defect due to abnormal vulval development, agar plates containing 200 μM FUDR were used to during the first seven days of adulthood to avoid internal hatching.

**Antibody staining.** Antibody staining of dissected gonads was performed with day-one adult animals[29]. A 1:200 dilution of a monoclonal mouse anti-GFP antibody (Roche, Nr. 11814460001) and a 1:25 dilution of the secondary antibody were used to detect ENDU-2::EGFP in the gonad.

***oma-1* RNAi inheritance assay.** *oma-1* RNAi was initiated at L1 stage at 20 °C (restrictive temperature) and these animals were designated as P0. Six P0 day-one adults animals were transferred onto OP50 seeded agar plates and incubated at 20 °C for scoring the embryonic lethality over generations. Embryonic lethality was examined by transferring of about 100 embryos onto an OP50 seeded agar plate and quantifying the number of hatched animals after 48 h.

**Oil Red O staining to quantify content of body fat.** Worms were fixed in 500 μl 60% isopropanol. After removal of supernatant, 500 μl freshly filtered ORO working solution was added to stain worms at 25 °C in a wet chamber overnight. Animals were washed once with 1 ml 0.01% Triton-x100 containing M9 buffer and responded in 250 μl M9-Triton-x100 buffer. Stained worms can be stored at 4 °C for at least 1 month. For quantification, color images were recorded. The original images were spitted into red and blue channels and the level of Oil Red-O was quantified by determining the excess intensity in the red channel in comparison to the blue channel. The mean lipid content per worm was calculated as the total intensity within stained regions normalized by the area of the worm regions. Oil Red O working solution was prepared as follows: 0.5 g of Oil Red O powder was

dissolved in 100 ml isopropanol solution and equilibrated for several days. The solution was then freshly diluted with 40% water for a 60% stock and allowed to sit overnight at room temperature and filtered using 0.2 mm filters.

**RIP-Seq**. To isolate ENDU-2 associated RNAs, animals of mixed stage were lysed in Lysis-NP-40 buffer (150 mM NaCl, 50 mM Tris pH 8.0, 1% NP-40 + Protease Inhibitor+100 μl/ml RNase Inhibitor) with SilentCrusher S($2 \times 30$ s $12,600 \times g$). Lysates were clarified by centrifuging at $16,000 \times g$ for 15 min. Supernatants were pre-cleared with Dynal Protein A magnetic beads (Invitrogen) and incubated with anti-GFP antibody (Abcam) for 1 h at 4 °C, followed by incubation with Dynal Protein A Magnetic Beads for 2 h at 4 °C. Beads were washed three times with Lysis-NP-40 Puffer before DNase I digestion (100 μl 1x DNase buffer, 1 μl DNase I, 1 μl RNase inhibitor) at 37 °C for 30 min. In all, 10 μl of 50 mM EDTA was added to stop DNase digestion. RNA was extracted with 5 volumes Trizol reagent (Invitrogen), followed by isopropanol precipitation. Precipitates were resolved in RNase-free water for visualization on 12.5% denaturing acrylamide gel with SYBR Gold (Thermo Scientific). Random hexamer primers were used for preparing TruSeq3 reverse-stranded cDNA libraries for Illumina single-end 50 bp sequencing. RIP-Seq to compare RNA-binding efficiency of ENDU-2(wt), ENDU-2 (E454Q), and ENDU-2(E460Q) at 15 and 25 °C were performed without cross-linking. RIP-Seq to determine RNA targets of ENDU-2 was carried out with ENDU-2(E454Q)::EGFP expressing animals raised at 15 °C with UV cross-linking.

**Analysis of RIP-Seq data**. The sequencing data were uploaded to the European Galaxy Server at https://usegalaxy.eu, and all read quality controls, trimming, mapping and counting were performed through Galaxy[30]. Specifically, after trimming with Trimmomatic 0.36[31], we mapped the reads to the *C. elegans* genome using the Wormbase WS260 genomic sequence (ftp://ftp.wormbase.org/pub/wormbase/releases/WS260/species/c_elegans/PRJNA13758/c_elegans.PRJNA13758.WS260.genomic.fa.gz) and the WS260 canonical gene set (ftp://ftp.wormbase.org/pub/wormbase/releases/WS260/species/c_elegans/PRJNA13758/c_elegans.PRJNA13758.WS260.canonical_geneset.gtf.gz) with the RNA aligner STAR 2.6.0b[32] with default settings, then counted reads mapping to individual genes using feature Counts 1.6.2[33]. For a read to be counted as mapping to a particular gene, we required a minimum read mapping quality of 12 (-Q 12 option of feature Counts) and an overlap of at least 1 base between the read and any of the exons of the gene (--minOverlap 1).

The IP enrichment of a transcript was expressed as FC = (IP RPM + 1)/(Mock IP RPM + 1), where RPM = number of reads mapped to a gene/(number of all mapped reads) $\times 10^6$. The addition of 1 in the FC formula prevents infinite enrichment. ENDU-2 associated RNAs were defined as transcripts with both FC > = 4 and RPKM > = 1 in ENDU-2(E454Q) IP sample. RPKM is calculated as follows: RPKM = RPM/length of transcripts $\times 10^3$.

**RNA-seq and data processing of isolated gonads**. Animals were raised at 25 °C from the L1 larval stage and the complete gonads were isolated 24 h after mid-L4 stage. Total RNA of the gonads was extracted with RNeasy® Mini Kit (Qiagen, Venlo, The Netherlands). Purification of poly-A containing RNA molecules, RNA fragmentation, strand-specific random primed cDNA library preparation, and single-read sequencing (50 bp) on an Illumina HiSeq 4000 were carried out by Eurofins Genomics. The RNA-seq results from two independent biological replicate of dissected gonads were uploaded to the European Galaxy Server at https://usegalaxy.eu. All read quality controls, trimming, mapping and counting were performed through Galaxy[30] using protocols like described for RIP-Seq (above). The DESeq2 (Galaxy Version 2.11.40.6 + galaxy1) was used to determine differentially expressed features from count tables of differential transcript abundances. The RPKMs throughout this study were normalized separately. Only the longest isoform of each gene was used as length of a transcript. To estimate genes with enriched expression in the soma, RNA-seq data from wild type animals raised at 25 °C from a previous study[20] were compared with our transcriptome from isolated gonads of wild type animals. The soma enrichment factor (SEF) was calculated with the formal SEF = (RPKM$_{whole\ animal}$ + 2)/(RPKM$_{gonad}$ + 2)]. Addition of factor 2 to the formal minimize large SEF value caused by low RPKM values.

**In vitro RNA-binding assay of recombinant ENDU-2 protein**. The RNA-binding assay was performed as described[34]. *fat-7* and *trcs-1* mRNA were transcribed from complementary DNA (cDNA) with biotinylated uracil. The presence and purity of RNA were examined on an RNA gel. This biotin-labeled RNA was incubated with recombinant ENDU-2(wt)::EGFP::3xFlag, ENDU-2(E454Q)::EGFP::3xFlag, and ENDU-2(E460Q)::EGFP::3xFlag proteins expressed in HEK 293T cells. The labeled RNA was captured and isolated using streptavidin beads. Proteins binding the RNA were then visualized via western blot. The uncropped blots are included in the Supplementary Fig. 11.

**RNA-cleavage assay**. Pellets of animals raised at 15 °C in mixed stages were washed twice with 1 ml 10 mM EGTA containing M9 buffer before resuspension in 0.7 ml NP-40 buffer. 3x Protease inhibitor was added to the buffer and worms were lysed with SilentCrusher ($2 \times 30$, $12,600 \times g$). 5 μl RNAsin Plus RNAse Inhibitor (40 U/μl) was added to the worm lysate and centrifuged by $15,000 \times g$ for 15 min at

4 °C. For the RNA-cleavage assay 95 μl supernatant of the worm lysate with 200 μg total protein was incubated with 5 mM $Ca^{2+}$, $Mg^{2+}$, $Mn^{2+}$, or 10 mM EGTA for 30 min at 7 °C before total RNA was extracted with Trizol. 0.2 μg RNA was loaded on 10% 8 M Urea acrylamide gel with SYBR Gold for visualization of RNA.

**Microarray**. Both *endu-2(tm4977)* and *endu-2(tm4977);byEx1375[endu-2P::endu-2::EGFP;myo-2::mCherry]* animals used for microarray were granddaughter decedents of one single *endu-2(tm4977);byEx1375[endu-2P::endu-2::EGFP;myo-2::mCherry]* animals (stain preparation see Supplementary Fig. 8). Animals were raised at 25 °C for 48 h after hatching and total RNA was purified by using RNeasy® Mini Kit (Qiagen, Venlo, The Netherlands). The following data analysis were performed with Partek Genomics Suite.

**GO term enrichment analysis**. GO term and tissue enrichment analysis was carried out with the online enrichment analysis tool (https://wormbase.org/tools/enrichment/tea/tea.cgi)[35].

**Quantification of mRNA level via quantitative RT-PCR (qPCR)**. Total RNA was prepared from day-one adult animals raised at 25 or 20 °C by using RNeasy® Mini Kit (Qiagen, Venlo, The Netherlands). *endu-2(tm4977)* animals were granddaughter decedents of *endu-2(tm4977);byEx1375[endu-2P::endu-2::EGFP;myo-2::mCherry]* animals. *act-4* was used as internal control. qPCR primer sequences: *cav-1* forward aagtgctggtggagtagatgc, *cav-1* reverse tccgatagcgatgttctcttc, *clec-169* forward tggacacttgtaactgtgcaaga, *clec-169* reverse ttttcatttgaccactttagatcg, *F38B6.4* forward cccttgtcctcggaaattaga, *F38B6.4* reverse caggcccgaagatggtaat, *pfas-1* forward caagattggaagttccgaaga, *pfas-1* reverse ctcaatatccggacagtcgtc, *B0286.3* forward gacccgaaaaacttggagctt, *B0286.3* reverse Tggaacgaatgagcacataatc, *atic-1* forward tgctacaaaaatgccagcag, *atic-1* reverse cgtttaaaggaagaccaacagc, *acox-1.2* forward agcggtgatctatggaagtga, *acox-1.2* reverse agctcagggatcttggacac, *abdh-3.2* forward atgacaccaccccaattgtc, *abdh-3.2* reverse tgctgtcatgagtacttcctgtg, *col-176* forward ccacaacctttgctccaatc, *col-176* reverse gagcacacttgatgcagtcg, *clec-15* forward tgcgccagaagggtattact, *clec-15* reverse cgcagaacgatctaaccaga, *asp-14* forward gctgcagttaccaacattacca, *asp-14* reverse agcaacgaagaaggttgagg.

**Single-molecule fluorescent in situ hybridization (smFISH) and data analysis**. In total, 38–48 probes of 20 nt in length targeting each mature mRNA were designed with Stellaris RNA FISH probe designer on gene target specificity. Probes were produced, conjugated to CAL Fluor Red 610, and purified by Biocat (sequences of the probe sets are available in Supplementary Data 4). Wild type and *endu-2(tm4977)* L4 animals were selected and dissected 24 h later for smFISH staining using a published protocol procedure[36]. Images were acquired with an Image Z1 fluorescence microscope. Exposure times and acquisition settings were identical between replicates. mRNA puncta were quantified by using ImageJ 1.51 s cell counter plug-in. Regions of interest for acquisition were defined by nuclei DAPI staining.

**Statistical analysis**. Experiments shown in this study were performed independently 2–4 times. Details of the particular statistical analyses used, precise *P*-values, statistical significance, number of biological replicas, and sample sizes for all of the graphs are indicated in the figures or figure legends. *n* represents the number of animals tested, unless mentioned otherwise. *N* means the number of biological replicates. \*\*\**P*-value <0.0001.

**Reporting summary**. Further information on research design is available in the Nature Research Reporting Summary linked to this article.

## Data availability

The authors declare that all data supporting the findings of this study are available within the article and its supplementary information files or from the corresponding author upon reasonable request. New raw data: Microarray data have been deposited in the ArrayExpresss database under accession code E-MTAB-7993. RIP-seq data have been deposited in BioProject under BioProject ID PRJNA544611. RNA-seq data of isolated gonad have been deposited in BioProject under ID PRJNA655512. Published data used in this study: The whole animal RNA-seq data[20] are downloaded from Gene Expression Ominous under accession code: GSE101524. The analyzed microarray, RIP-Seq and RNA-seq data of isolated gonad are included in Supplementary Data 1–3.

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

## Acknowledgements
We thank the Caenorhabditis elegans Genetics Center which is funded by the NIH Office of Research Infrastructure Programs (P40 OD010440) for providing some strains; Dr. S Mitani for providing endu-2(tm4977) strain, Cornelia Habacher for sharing the RIP-Seq protocol, D. Sherwood for providing the fos-1 promoter. We thank the staff of the Life Imaging Center (LIC) of the Albert-Ludwigs-University Freiburg for their microscopy resources. This work was funded by grants from the German Research Foundation (DFG) (SFB850, SFB1381), Germany's Excellence strategy (CIBSS-EXC-2189 - Project ID 8 390939984) and the German Excellence Initiative (BIOSS - EXC294) to R.B.

## Author contributions
W.Q. and R.B. designed the experiments and analyzed data. W.Q., E.D.v.G., F.X., Q.Z., L.L., and W.Y. performed the experiments. D.P. performed and analyzed microarray. W. Q. and W.M. analyzed RNA-Seq data. W.Q. and R.B. wrote the manuscript.

## Funding

## Competing interests
The authors declare no competing interests.
