## [Peer Review File · Nature Communications]

Reviewers' Comments:

Reviewer #2:

Remarks to the Author:

The authors have addressed my major concerns and I recommend publishing the work in *Nature Communications*. In particular, new data regarding the non-autonomous role of *ENDU-2* in promoting germline immortality seem of high quality and do a good job addressing my major original concerns. RNA profiling analyses were not a strong-point to me, but seem acceptable for publication. Writing is very clear and logical and I enjoyed reading the revised paper quite a bit. The revised work still needs some editing for grammar.

Reviewer #4:

Remarks to the Author:

I have systematically looked over all of points raised by Reviewer 3 and I tried to sum up if and how the authors addressed those concerns in the paragraphs that follow.

Point 1-The entire section that caused the problem for the reviewer was removed and apparently incorporated into another study.

Point 2-

A) The reviewer also had comments about the inaccurate measurements of protein in the embryos that had been used to show temperature differences in *ENDU-2* levels. These experiments were not re done and instead the authors elected to remove the claim that *ENDU-2* levels are regulated by temperature. They do not claim this now and there is no appended western blot.

B) the initial micrographs apparently showed that GFP from *ENDU-2* were still visible in the extracellular space despite the lack of a ss on the fusion protein. This was apparently due to fluorescence emanating from the gut and in a new version of the figure the authors show both accumulation and lack of accumulation w or w/o a ss and restoration using a heterologous ss. This is quite convincing, particularly the use of the *sel-1* ss.

C) The reviewer made a point that additional experiments would be important to reinforce the claim that *endu-2* affects the *Mrt* phenotype in a temperature dependent manner by performing RNAi experiments in both an *rrf-1* or *ppw-1* background to isolate the RNAi effect to the germline. The authors claim that such experiments were indeed performed, but had not shown any effect and suggest that this likely results from ineffective RNAi in the neurons. However, the authors claim that the major contribution of *endu-2* is intestinal and not from the neurons, and they go to some extent to distinguish this from the original paper by Ujisawa et al. that shows that a neuronal *endu-2* activity is important for the non-autonomous effect on cold tolerance. This still seems unresolved to me and seems like an inadequate explanation to account for these findings, particularly since there are several sensitized backgrounds that enhance neuronal RNAi. The authors do however try to address this concern by performing tissue specific transgene expression that convincingly shows that only expression of *endu-2* in the gut or the neurons is sufficient to correct the progressive *Mrt* phenotype that is observed in the *endu-2* mutants at higher temperatures.

D) The claims about the effects of *endu-2* on lifespan that were part of a previous version of the manuscript were removed as they appeared to cause some contention.

Point 3

A,B) The reviewer had significant issues with the method used for the analysis and the annotation/representation of the data. Most of these concerns were addressed in a new RIP-seq analysis and representation in Figure 4. These included a more detailed description of the methods employed to ensure that the experiments and the claims made by the authors could be reproduced

and verified.

C) The reviewer was unsure why the authors used a transcript length normalization instead of transcript numbers and was somewhat addressed in the new figure.

D) Finally the authors' claim that ENDU-2 preferentially bound processed transcripts was supported by RIP-seq data that indicated that most RNA targets were aligned to exon sequence and not introns.

Point 4

The reviewer was concerned with the overall interpretation that the effect of ENDU-2 was to repress somatic gene expression in the germ line. This was difficult to refute apparently with the previous data, but the authors carried out a series of additional experiments including RNA-seq analysis on isolated gonads. These data are consistent with somatic targets being repressed in an ENDU-2-dependent manner in the germ line albeit it is not extensively validated experimentally beyond the comparison of the datasets described in the response to reviewers. It is a bit surprising that ENDU-2 is capable of making this distinction but there must be something specific about the germ line RNAs that spares them from ENDU-2 binding. Perhaps additional germline-specific RNA analysis as performed in Figure 4C would make this clearer. Presumably, germline mRNAs would not be precipitated with ENDU-2 based on the interpretation of the authors. Regardless, a characteristic signature must be present on germline-specific transcripts that would impede ENDU-2 binding, and so far I am not aware of any such structure. Alternatively, ENDU-2 would have to function during or prior to RNA synthesis at somatic loci (such as via *hda-1* as indicated by the Jia et al. paper) that are being expressed in the germ line. Nevertheless, it is not at all clear how this enzyme carries out its final purported effect once it leaves the soma and affects its targets in the germ line.

The rest of the points that were brought up by the reviewer were addressed by removing the corresponding experiments or section of the manuscript leaving only a subset of experiments that were initially part of the original submission. The numerous overinterpretations and overstatements that were present in the original submission were mostly diluted or removed by the authors and the revised version is largely based on relatively sound conclusions drawn from convincing data.

The rest of the points made by the reviewer were essentially editorial in nature and were more or less addressed by the authors. Despite these improvements however there are still a number of typos and grammatical errors throughout the manuscript.

From my perspective, the authors acted in good faith and have addressed most of the major concerns of the previous reviewer, albeit in doing so the manuscript is probably now only a subset of the original findings that initially put it into review. There are still some issues that I just do not understand and the findings presented here do not clarify them at all.

Among the most curious issues is the final interpretation that ENDU-2 affects the expression of somatic transcripts. How might it do this? Could ENDU-2 bind to a somatic nuclear Argonaute complex that is aberrantly active in heat affected germ cells? There is no model or evidence to indicate how ENDU-2 might achieve this specificity for its various RNA targets.

Furthermore, how do the authors account for the progressive transgenerational Mrt phenotype? That *endu-2* mutants may be progressively sterile in higher temperatures could be mediated through some aspects of its interaction with *ctps-1*, which is described in the Jia et al paper in 2020. These data are not discussed at all in the light of this phenotype despite that the sterility and reduced number of germ cells seem to be common to both situations. The data presented here are more consistent with a binary decision to block or not to block somatic gene expression in the germ line, however the observation that the Mrt phenotype occurs over multiple generations is not indicative of a simple toggle. How does the loss of this protein affect the germ line only after multiple generations of somatic gene expression? Is there a threshold level of somatic gene expression that is tolerated? Could it actually be associated with a single gene target/transcript? A number of papers have been published in the last few years that have described similar contexts of cell non-autonomous regulation of the germ line, both intergenerationally and transgenerationally. Many of these have provided more questions than answers. In this case, the

experiments are sound and, at least in the present version, the conclusions are more consistent with the results that are presented. The work extends the previous finding by Ujisawa et al. that ENDU-2 can act non-autonomously, and can affect numerous processes, including reproduction. The authors in the current study show that indeed it is secreted and it affects germ line gene expression by an unknown mechanism that has nothing to do with its ability to degrade RNA. Despite these interesting findings, the manuscript falls short of answering many of the questions that it raises.

Reviewer #2 (Remarks to the Author):

The authors have addressed my major concerns and I recommend publishing the work in Nature Communications. In particular, new data regarding the non-autonomous role of ENDU-2 in promoting germline immortality seem of high quality and do a good job addressing my major original concerns. RNA profiling analyses were not a strong-point to me, but seem acceptable for publication. Writing is very clear and logical and I enjoyed reading the revised paper quite a bit. The revised work still needs some editing for grammar.

We are happy to see that the Reviewer #2 positively evaluated our revised manuscript. We have asked colleagues to help with editing for grammar.

Reviewer #4 (Remarks to the Author):

We are happy to see that the reviewer #4 has systematically looked over all of points raised by Reviewer 3 and recognized that we had acted in good faith and addressed most of the major concerns of the previous reviewer.

Here are our responses to the additional questions raised by reviewer #4.

The claims about the effects of endu-2 on lifespan that were part of a previous version of the manuscript were removed as they appeared to cause some contention.

We decided to remove the lifespan data according to the suggestions from the other two reviewers and only focused on explaining how ENDU-2 ensures germline immortality. We are investigating the lifespan regulatory aspect of ENDU-2 currently and will submit these results in another manuscript when more molecular mechanistic details could be discovered.

The reviewer was concerned with the overall interpretation that the effect of ENDU-2 was to repress somatic gene expression in the germ line. This was difficult to refute apparently with the previous data, but the authors carried out a series of additional experiments including RNA-seq analysis on isolated gonads. These data are consistent with somatic targets being repressed in an ENDU-2-dependent manner in the germ line albeit is it not extensively validated experimentally beyond the comparison of the datasets described in the response to reviewers. It is a bit surprising that ENDU-2 is capable of making this distinction but there must be something specific about the germ line RNAs that spares them from ENDU-2 binding. Perhaps additional germline-specific RNA analysis as performed in Figure 4C would make this clearer. Presumably, germline mRNAs would not be precipitated with ENDU-2 based on the interpretation of the authors. Regardless, a characteristic signature must be present on germline-specific transcripts that would impede ENDU-2 binding, and so far I am not aware of any such structure. Alternatively, ENDU-2 would have to function during or prior to RNA synthesis at somatic loci (such as via *hda-1* as indicated by the Jia et al. paper) that are being expressed in the germ line. Nevertheless, it is not at all clear how this enzyme carries out its final purported effect once it leaves the soma and affects its targets in the germ line.

Our RIP-seq results suggest that a lot of germline mRNAs (e.g. *trcs-1*, *pgl-1*, *pgl-2* and *pgl-3* mRNA) are bound by ENDU-2 and the RNA binding affinity play an essential role in maintenance of germline immortality (Fig. 4 and Supplementary Data 1). The difference in the regulatory roles of ENDU-2 in the germline and in the soma is whether RNA cleavage activity is involved or not. Even though we don't have any data yet how ENDU-2 represses soma specific transcripts in the germline, we are agree with the opinion of this reviewer that ENDU-2 might act at the transcriptional level in the germline. And this will be one subject studied in the future.

The rest of the points that were brought up by the reviewer were addressed by removing the corresponding experiments or section of the manuscript leaving only a subset of experiments that were initially part of the original submission. The numerous overinterpretations and overstatements that were present in the original submission were mostly diluted or removed by the authors and the revised version is largely based on relatively sound conclusions drawn from convincing data. The rest of the points made by the reviewer were essentially editorial in nature and were more or less addressed by the authors. Despite these improvements however there are still a number of typos and grammatical errors throughout the manuscript.

We have asked colleagues to help with correcting typos and grammatical errors.

Among the most curious issues is the final interpretation that ENDU-2 affects the expression of somatic transcripts. How might it do this? Could ENDU-2 bind to a somatic nuclear Argonaute complex that is aberrantly active in heat affected germ cells? There is no model or evidence to indicate how ENDU-2 might achieve this specificity for its various RNA targets.

This is a very important question. Germ granules were shown to inhibit germline expression of soma specific genes¹. In addition, we observed that germline ENDU-2 was localized in punctate structures that could be P granules, mutator granules or Z granules. Therefore, ENDU-2 might regulate functions of the germ granules to prevent misexpression of the somatic transcripts. We have include this hypothesis in the discussion.

The high light of our current manuscript is that we have identified a conserved protein secreted from the soma and it can control gene expression in the germline. To understand how functional specificity is achieved requires a large amount of future work combined with biochemical, molecular biological and genetic assays. We are currently trying to identify proteins/protein complexes co-immunoprecipitated with ENDU-2 via LC-MS. This MS interactor screen will reveal whether nuclear Argonaute complexes are involved and which RNA binding proteins might team up with ENDU-2 to control gene expression in the respective tissues.

Furthermore, how do the authors account for the progressive transgenerational Mrt phenotype? That *endu-2* mutants may be progressively sterile in higher temperatures could be mediated through some aspects of its interaction with *ctps-1*, which is described in the Jia et al paper in 2020. These data are not discussed at all in the light of this phenotype despite that the sterility and reduced number of germ cells seem to be common to both situations. The data presented here are more consistent with a binary decision to block or not to block somatic gene expression in the germ line, however the observation that the Mrt phenotype occurs over multiple generations is not indicative of a simple toggle. How does the loss of this protein affect the germ line only after multiple generations of somatic gene expression? Is there a threshold level of somatic gene expression that is tolerated? Could it actually be associated with a single gene target/transcript?

Our data suggest that the progressive sterility in *endu-2(lf)* mutants is more likely caused by excessive cell death in the germline rather than defects in the germline proliferation (Fig. 1b and Supplementary Fig.2b). In contrast, *ctps-1* knock-down leads germless phenotype due to a proliferation defect². In addition, ENDU-2 must be secreted and taken up by the gonad to preserve germline immortality while ENDU-2 primarily controls CTPS-1 activity in the soma. Therefore, we consider that functional interaction with CPTS-1 is unlikely to be involved in maintenance of germline immortality. We have included this analysis in the Results section of the manuscript.

Why does the completely sterility only appear after several generations? It is possibly due to toleration of a threshold level of somatic gene expression, as suggested by this reviewer. Another possibility is that even loss of immortality abrogates the capacity of infinitive proliferation and maintenance of the stem cells, they could still undergo certain rounds of cell division until that cell death occurs. Therefore, sterility would only be observed after several generations.

References

- 1 Knutson, A. K., Egelhofer, T., Rechtsteiner, A. & Strome, S. Germ Granules Prevent Accumulation of Somatic Transcripts in the Adult *Caenorhabditis elegans* Germline. *Genetics* **206**, 163-178, doi:10.1534/genetics.116.198549 (2017).
- 2 Jia, F., Chi, C. & Han, M. Regulation of Nucleotide Metabolism and Germline Proliferation in Response to Nucleotide Imbalance and Genotoxic Stresses by EndoU Nuclease. *Cell Rep* **30**, 1848-1861 e1845, doi:10.1016/j.celrep.2020.01.050 (2020).